# Rearing of *Hermetia Illucens* on Different Organic By-Products: Influence on Growth, Waste Reduction, and Environmental Impact

**DOI:** 10.3390/ani9060289

**Published:** 2019-05-29

**Authors:** Luciana Bava, Costanza Jucker, Giulia Gislon, Daniela Lupi, Sara Savoldelli, Maddalena Zucali, Stefania Colombini

**Affiliations:** 1Dipartimento di Scienze Agrarie e Ambientali—Produzione, Territorio, Agroenergia, Università degli Studi di Milano, via Celoria 2, 20133 Milano, Italy; luciana.bava@unimi.it (L.B.); giulia.gislon@unimi.it (G.G.); stefania.colombini@unimi.it (S.C.); 2Dipartimento di Scienze per gli Alimenti, la Nutrizione, l’Ambiente, Università degli Studi di Milano, via Celoria 2, 20133 Milano, Italy; costanza.jucker@unimi.it (C.J.); daniela.lupi@unimi.it (D.L.); sara.savoldelli@unimi.it (S.C.)

**Keywords:** black soldier fly, LCA, animal feeding, larval development, bioconversion, by-products

## Abstract

**Simple Summary:**

In the last few years great efforts have been made to find alternative protein sources to soybean meal and valorize organic by-product, which are produced in large amounts by food industries and are often inedible by humans. In this context, insects could be an alternative protein source. The aims of this study were to evaluate the growth performance, nutrient composition, and environmental impact of *Hermetia illucens* reared on a control hen diet and by-products diets (okara, maize distillers, and brewer’s grains). The rearing substrate affected larvae growth performance, nutritive value, and environmental impacts. Larvae fed on a hen diet and the maize distiller exhibited the higher final weight and required fewer days to reach the prepupal stage. The lipid content of the larvae was positively correlated to that of the insect diets. The environmental impact of larvae production on the hen diet, characterized by a high inclusion of soybean meal, was the most impactful for most of the environmental categories. Feed production activities were the main contributor to environmental impact. The use of by-products for larvae growth allowed us to reduce the environmental impact to produce 1 kg of protein in comparison with the use of soybean meal as a protein source.

**Abstract:**

The aim of the study was to evaluate the use of three by-products as growing substrates for *Hermetia illucens* (Black Soldier Fly (BSF)) larvae: okara, maize distiller, brewer’s grains, and a control hen diet. The study focused on larval growth and bioconversion performance, production of methane by larvae and environmental burden of larvae production, using Life Cycle Assessment (LCA) on a lab scale. Chemical composition of substrates differed: okara had the highest crude protein and ether extract contents, while brewer’s grains showed the highest fiber content. Larvae fed on a hen diet and maize distiller exhibited the highest final weights (2.29 and 1.97 g, respectively). Larvae grown on okara showed the highest indexes for waste reduction and efficiency of conversion of the ingested feed. The BSF larvae did not produce any detectable traces of CH_4_. LCA evaluation showed that larvae production on a hen diet resulted in the most impact for most of environmental categories, for the inclusion of soybean meal in the diet (for climate change, 5.79 kg CO_2_ eq/kg dry larvae). Feed production activities resulted in the main contributions to environmental impact. In order to compare the larvae production obtained on all substrates, an environmental impact was attributed to okara and brewer’s grain through a substitution method, and, by this approach, the best sustainable product resulted from the larvae grown on the maize distiller.

## 1. Introduction

By-products are incidental or secondary products resulting from a production process, whose main purpose is not the production of the item itself. Most by-products derive from the agri-food industry and are largely used for livestock feeding due to their nutritive value. The use of by-products as feed affords considerable advantages, such as the reduction of waste production, a reduction in competition between animals and human for crops, and the possible reduction of feeding costs. A possible alternative exploitation of by-products is represented by their use as a rearing substrate for insects, constituting an interesting example of a sustainable circular economy [1,2,3]. In this context, organic by-products can be valorized to produce a valuable insect biomass, which is rich in protein and fat, for the animal feed industry [4,5,6] or biodiesel production [7,8,9]. Among the most widely available by-products from the food industry, some show interesting characteristics, such as high protein and energy content, useful to satisfy the insects’ nutritive requirements. Among these, okara (soy pulp or fiber obtained from tofu production), wet brewers’ spent grains (from the beer industry), and maize distiller’s grains (from spirit or ethanol production) represent a possible substrate for insect rearing. Okara availability has been increasing throughout the world due to an increased production of soybeans, and the dumping of okara has become a problem due to its impact on the environment [10]. Similarly, brewer’s spent grain forms up to 85% of the by-products of the brewing industry [11], and in the EU, about 3.4 million tons of brewers’ spent grains are generated annually in beer production [12]. 

Larvae of *Hermetia illucens* (Diptera: Stratiomyidae), also known as the black soldier fly (BSF), can grow on a wide range of organic material, ranging from fruits and vegetables to animal manure [1,13,14,15]. Adults are able to survive without feeding, but their lifespans can be significantly extended in the presence of a sucrose solution [16,17]. From previous studies, larvae have had an average content of 45.2% crude protein and 31.4% fat. However, fat and protein composition can vary as a function of growing substrate characteristics [5,14,15]. Another advantage of BSF is that, unlike many pests that consume waste, BSF do not carry bacteria or diseases and larvae are capable of inactivating *Escherichia coli* and *Salmonella enterica* [18]. 

Recent studies, which evaluated by-products as substrates for insect growth [19,20,21], underlined that the environmental load of feed takes more than half of the impact of insect production. From a Life Cycle Assessment (LCA) point of view, the inclusion of a by-product, often without economic value, in another production process could cause some methodological problems. Thus, there is not scientific consensus for assigning an environmental load to by-products. In many studies, by-products or food waste assume zero burden [21,22], as the authors considered only the environmental load of the transport of by-products or waste in the secondary process. On the other hand, if residues are continuously considered burden-free by default, materials (by-products) from systems that cause a high environmental impact can be superior for their environmental performance when compared to other resources [23]. The problem could be solved through an allocation strategy for environmental impact between the principle products and by-products based on mass, or through an economic or biophysical approach, but this approach is not always possible for food by-products. For example, by-products often do not have an economic value, or their mass is higher than the primary products (i.e., trub from beer production). Another possibility suggested by many authors [23,24,25] is the substitution of by-products with other comparable products. This solution was applied by Eriksson et al. [24], where vegetable waste was assumed to replace oats (based on energy content) in feed pig diets.

The aim of the study was to exploit three inedible human by-products as growing substrates for BSF larvae. Okara, maize distiller and brewers’ spent grains mixed with trub (later called brewer’s grain) were tested as rearing substrates for BSF production at an experimental scale. The by-products were chosen based on availability in the territory, ease of retrieval, water content, and cost. The study focused on (1) larval growth performance, bio-reduction of the growing substrate and production of methane by BSF larvae; and (2) environmental burden, with an LCA approach, of larvae production grown on tested substrates, including the environmental impact of by-products though an alternative approach.

## 2. Materials and Methods 

### 2.1. Growth Performance

Larvae used in the current study were provided from a BSF colony previously founded by wild specimens collected in Lombardy (Northern Italy) (45°19’54’’N; 9°05’58’’E) [2]. The following rearing substrates were tested: okara, maize distiller, brewer’s grains, and a hen diet. The chosen substrates were available in the area where the experiment was performed, while the hen diet was chosen as a control. For each experimental diet, 1000 2-day-old larvae were placed in a plastic container (21 × 27 × 16 cm) covered with a mesh netting and stored in climate chamber (Temperature 25 ± 0.5 °C, Relative Humidity 60 ± 0.5%, photoperiod 12:12 L:D). Substrates were tested in triplicates. The hen diet and maize distiller diets were added with water (1:1 volume). Each diet was provided ad libitum, water was added when necessary, and the total amount was measured. In order to study larval growth performance, ten larvae of each replicate were weighed every three days with an analytical balance (SartoriusCP64, Germany). Each trial ended when 40% of the larvae in each container reached prepupal stage (indicated by the darker color of the larvae) [26]. Thus, prepupae were removed daily from the container and counted. At the end of each trial, the number of living prepupae was registered to evaluate larval survival. In order to assess the CH_4_ production, twenty BSF larvae with a weight between 80 and 100 mg were placed in duplicate into serum bottles with the addition, ad libitum, of the experimental diets. For each sample two blanks (i.e., substrates without larvae) were added. After 24 h of incubation, a fixed volume of gas was collected for subsequent methane analysis using gas-tight syringes fitted with needles through the bottle top. The CH_4_ concentration of the headspace was determined by micro gas chromatograph (Agilent Technologies, Santa Clara, CA, USA). 

The method suggested by Diener et al. [27] was used to calculate the larval growth rate (g d-1) as follows:Larval growth rate (GR) = (final larval average weight − initial larval average weight)/number of days of the trial.

#### Conversion Efficiency

To evaluate larval efficiency in consuming and metabolizing the growing substrates, the total final biomass (larvae + pupae) and the residual substrates were weighted. Waste reduction index (WRI) and the efficiency of conversion of the ingested feed (ECD) were calculated for the determination of the waste consumed by the larvae and the conversion efficiency of the substrates into valuable biomass. The following indexes, based on dry weight, were calculated, as in Diener et al. [27]: Waste reduction index (WRI) = (W − R/W)/days of trial (d) × 100where W = total amount of diet provided; R = residual of the diet, and
Efficiency of conversion of the ingested food (ECD) = B/(W−R)
where B = total larval + pupal biomass (g); W = total amount of diet provided; R = residual of the diet.

### 2.2. Substrate Composition and Larvae Composition Analysis

Larvae and substrate samples were freeze dried. After freeze-drying, the substrates were ground through a 1 mm screen (Pulverisette 19, Fritsch) while larvae were ground by pestle. Larvae were analysed for concentrations of dry matter (DM) (method 945.15; [28]), ash (method 942.05; [28]), crude protein (CP) (method 984.13; [28]), and ether extract (EE) (method 920.29; [29]). Substrates were also analysed for neutral detergent fiber (NDF) with the addition of α-amylase according to Mertens [30] and using the Ankom 200 fiber apparatus (Ankom Technology Corp., Fairport, NY). Non fibrous carbohydrates (NFC) were calculated as follows: NFC= 100 − Ash − CP − EE − NDF

### 2.3. Environmental Impact Assessment

#### 2.3.1. Goal and Scope Definition, Functional Unit Selection

The environmental impact of *H. illucens* larvae production on the three different organic by-products, and on a hen diet, was evaluated using a life cycle assessment method. Maize distiller grain, brewer’s grains, and okara were obtained directly from the food industry, while the hen diet was purchased from the market. Since larvae can be used as feed for livestock, substituting other high impactful raw materials, such as soybean, different functional units (FU) were used in order to allow a comparison with the other feeds. The FUs used were 1 kg of larvae (dry wet weight), 1 kg of protein from larvae, and 1 kg of fat content from larvae, expressed as ether extract. 

#### 2.3.2. System Boundaries

The system boundaries (Figure 1) included the development of BSF from eggs to prepupae. Energy, transport, water, and substrate production were considered. Larvae manure and feed residues obtained after separation of larvae from the rearing substrate were analysed for N, K, and P composition. For the N analysis method, 984.13 [28] was applied. The total P and K contents were determined by inductively coupled plasma mass spectrometry (Varian, Fort Collins, CO, USA), preceded by acid digestion [31] of the samples. Manure was considered to be avoided fertilizer in the environmental impact calculation.

#### 2.3.3. Inventory Data Collection

Primary data on insect growth, feed consumption and water consumption were obtained directly from the experimental rearing trials and they are showed in Table 1. 

The background data and emissions related to substrates were considered as follows:For a hen diet, considering all the single ingredients of the diet, the majority of ingredients came from European countries and the emissions were quantified using data from Ecoinvent V3 and Agri-footprint [32] databases. Protein feeds, mainly soybean meal, originated from Brazil. Direct land use change (LUC) for soybean meal and soybean oil productions were considered in the assessment using the value reported by the Agri-footprint database (Soybean, at farm/BR Economic, [32]).For maize distilled from ethanol, the value proposed by the Ecoinvent V3 database was used with economic allocation (2.4%). It was assumed that it had been produced in Italy (50%) and Germany (50%).For brewer’s grains mixed with trub and okara, zero environmental impact was assumed because they did not actually have economic value. In order to analyse the possible effects of different allocation choices on by-product environmental impacts assessment, a sensitivity analysis was performed, as explained below.A comparison of different larvae with other important animal feed types with high protein content (fish meal and sunflower meal) and high fat content (vegetable oil and rapeseed meal) were performed. The background data and environmental impacts of these feeds were obtained from the following databases: LCA Food DK for fish meal, Agri-footprint for sunflower meal, and Ecoinvent for vegetable oil and rapeseed meal.

#### 2.3.4. Impact Assessment 

Within the LCA, SimaPro 8 software [33] was used to estimate the environmental impact of the tested systems. The following impact categories were considered for evaluation: climate change, ozone depletion, particulate matter, photochemical ozone formation, acidification, terrestrial eutrophication, freshwater eutrophication, marine eutrophication, land use, water resource depletion, and mineral and fossil renewable resource depletion. The characterization factors considered were those from ILCD (International Reference Life Cycle Data) 2011 Midpoint V1.03. The ILCD 2011 midpoint method was realized by European Commission, Joint Research Center. The method includes 16 midpoint impact categories. In the present study the most considerable categories were selected.

#### 2.3.5. Sensitivity Analysis

Sensitivity analysis was performed to investigate the variability of environmental impacts of larvae production due to the variation of the environmental impact associated with the two by-product substrates (brewer’s grains and okara) that do not have an environmental load evaluation in the main databases. Two approaches were used:(1)The substitution of the whole amount of each by-product with soybean meal, which is the main protein feed for animal production.(2)The substitution of the whole amount of each by-product with sunflower meal, which is a protein feed source produced in the European area.

### 2.4. Statistical Analysis

Statistical analyses were performed by SPSS^®^ statistic analysis software [34]. Data recorded on the time of larval development, final larval weight, larval survival, and conversion efficiency of BSF on the four substrates were compared by one-way analysis of variance (ANOVA). Tukey-Kramer’s Honestly Significant Difference (HSD) multiple comparison test was applied for the mean separation (*p* < 0.05) between the tested diets, where significant differences occurred.

## 3. Results

### 3.1. Substrate Chemical Composition, Performance Of Larval Growth and Bio-Reduction of the Substrates

Chemical composition of the experimental substrates is reported in Table 2. Substrates were characterized by a wide variability in chemical composition. The hen diet had the highest value of ash (13.5% on DM) as compared to the other diets (on average 4.55% on DM). A wide variability for CP and EE concentrations was also observed. Particularly, okara showed the highest values of CP (39.2% on DM) and EE (17.2% on DM) as compared to the other substrates, while hen and brewer’s grains diets were characterized by the lowest concentrations of CP and EE. The brewer’s grain substrate was also characterized by the highest NDF fiber content (53.6%) and the hen diet was characterized by the lowest NDF fiber content (15.7% on DM). BSF larvae did not produce any detectable trace of CH_4_.

In Table 3, the effect of the rearing substrates on the larval development, survival, and conversion efficiency are reported. All the tested substrates allowed for BSF larval growth and development, but the diet influenced the final weight of larvae. Larvae fed on a hen diet and maize distiller showed the highest final weight (2.29 ± 0.20 and 1.97 ± 0.14 g/10 larvae, respectively), and were statistically different from okara and brewer’s grain (F-Fisher = 21.85; degrees of freedom = 3, 8; *p* < 0.05). Moreover, larvae grown on the maize distiller and hen diet required fewer days to reach the prepupal stage (16 ± 0.58 and 15 ± 0.58 days, respectively). However, only larvae reared on brewer’s grains significantly differed from the others and required more days to reach the prepupal stage (22 ± 0.58 days) (F = 29.00; dF = 3, 8; *p* < 0.05). The weight gained per day (gd-1) was the highest when the larvae were fed with the maize distiller (0.0056 ± 0.0001 g d-1), followed by larvae on the control diet. Larvae grown on okara and brewer’s grains showed a smaller weight increment—significantly lower than the larvae on the other two substrates. Larval mortality was not affected by the rearing substrates (F = 4.005; dF = 3, 8; *p* > 0.05). All by-products allowed a high survival rate, with a total mean of 91.2 ± 4.11%. 

In order to acquire data on the BSF larval capability to reduce the waste and the efficiency to convert the diet into biomass, the WRI and ECD indexes were calculated. Statistical differences were observed for both indexes depending on the different rearing substrates (WRI: F = 37.32; df = 3, 8; *p* < 0.05; ECD: F = 7.76; df = 3, 8; *p* < 0.05). As shown in Table 3, WRI indexes ranged from a minimum of 3.01 ± 0.06 for brewer’s grain larvae to a maximum of 4.90 ± 0.07 for okara larvae. Larvae reared on maize distiller and brewer’s grain, the substrates with the highest NDF content (especially brewer’s grain), were statistically different from the okara and hen diet, with a lower capability to reduce the substrates. Considering the ECD index, only larvae grown on okara differed significantly from the others, showing a higher index (0.36 ± 0.02), thereby corresponding to a major attitude to convert the ingested feed. 

### 3.2. Larvae Chemical Composition

Larvae chemical composition in terms of ash, CP, and EE is reported in Table 4. The average concentrations of ash, CP, and EE were: 7.50%, 52.9%, and 27.4% on DM. Rearing substrates affected ash and EE content. In particular, ash content was higher for larvae grown on a hen diet (11.7%) compared to other treatments (on average 6.09% on DM). On the other hand, ether extract was higher for larvae grown on okara and maize distillers compared to hen and brewer’s grains diets. 

### 3.3. Environmental Impacts Assessment

The chemical characterization of feed refusal and manure, in terms of N, P, K content, is shown in Table 5. These analyses were done to assess the possible use as fertilizer of feed refusal and manure produced during larvae growth. The highest N and P contents were found using the maize distiller, while the highest K content was found using okara as a growing diet. 

The environmental impact evaluation for the production of 1 kg of dry larvae is shown in Table 6. In these results, no environmental impact was attributed to the productions of okara and brewer’s grains. The environmental impact of larvae production on the hen diet resulted in the most impact for most of the environmental categories, such as climate change, acidification, eutrophication (terrestrial and marine), and land use. The main gas contributor to climate change caused by larvae production using a hen diet was CO_2_ due to land transformation (51%), which was caused by the inclusion of soybean meal in the diet; the second highest (39%) was CO_2_, which was used as an input for the crop production (e.g., maize) or feed processing (drying and milling).

Figure 2 and Figure 3 show the weight of the different processes that contributed to the environmental impact of larvae production expressed as one kilogram of dry larvae. The results are reported only for the hen diet and maize distiller systems, because, unlike the other two systems, in these systems, an environmental weight has been assigned to the substrate used. 

For both systems, feed production activities resulted in the main contribution to environmental impact (from 60% to 99% of the impact depending of the different impact categories and substrates).

Considering that all over the world larvae production is now studied as an alternative feed and/or food, it is useful to evaluate the environmental impact of using it as a functional unit protein and for its fat content. Figure 4 compares climate change for the production of 1 kg of protein derived from (1) BSF larvae grown on the experimental substrates, (2) fish meal, and (3) two vegetable feed sources (sunflower and soybean meal). Sunflower and soybean meal are two by-products largely used as feed in animal production systems (both monogastrics and ruminants), due to their high crude protein content; their environmental impact is allocated between oil and meal, usually following an economic allocation.

Considering protein production, the lowest value was attributed to fish meal; protein production from larvae grown on maize distiller resulted comparably to the climate change for the production of 1 kg of protein from sunflower meal (3.76 and 3.80 kg CO_2_ eq/kg protein, for BSF on maize distillers and sunflower meal, respectively). 

After analysing the climate change generated for the production of 1 kg of lipid (Figure 5), the results suggest that the production of 1 kg of lipids is generally more impactful than the production of 1 kg of protein, mainly because the feed used for animal diets are often by-products, and the major quantity of fat is extracted into the main product, the vegetable oil, used for human nutrition, or biodiesel. The comparison shows that lipid production from BSF achieved the worst result, particularly when larvae were grown on a hen diet.

In the present study, climate change and marine eutrophication (per kg of lipid) was higher when using insects grown on hen diet than rapeseed meal (0.145 vs 0.035 kg N eq for BSF on hen diet and rapeseed meal for marine eutrophication, respectively), while marine eutrophication produced by BSF grown on maize distiller was the lowest (0.008 kg N eq). Significant benefits were found to be connected to acidification using BSF grown on the maize distiller as a source of lipids (0.015; 0.036; 0.008; 0.19 molc H^+^ eq, respectively, for the vegetable oil mix, rapeseed meal, BSF on maize distillers, and BSF on the hen diet).

### 3.4. Sensitive Analyses

The allocation choices for by-products have important effects on their environmental load and the ‘secondary’ systems associated to them, such as insect production. In the present study, two substituting approaches were used in order to assign an environmental load to okara and brewer’s grains. Substituting the quantity of substrates used in each system with a quantity of other protein feeds, on the basis of protein content, is suggested by Olofsson and Borjesson [23]. Thus, an alternative environmental load for the BSF larvae production, grown on the two by-products, was assigned. In this study, two protein feed were used: soybean meal and sunflower meal.

As shown in Figure 6, climate change and water resource depletion became higher when soybean meal was used as a substrate substitution. This result is due to the LUC load for the climate change category and the high water requirement compared to sunflower crop production. For the marine eutrophication and acidification categories, sunflower meal substitution was more impactful due to ammonia and nitrate emission during crop production.

Moreover, the results obtained from the sensitivity analysis, compared to the previous results, showed that the environmental impact and, in particular, the climate change of the larvae grown on okara and brewer’s grain (as substitutions for soybean meal) was higher than the impact of larvae grown on a hen diet and maize distillers. Similar or lower results were obtained for BSF on the two by-products (in each substitution) for marine eutrophication and water resource depletion. 

## 4. Discussion

This study aimed to evaluate the growth performances and environmental impact of BSFs reared on different by-products on a lab scale. The results of the present study, although not derived from an industrial producing process, provide estimates on possible sustainable uses of insects for feed, considering different rearing substrates. Hence, the results provide significantly reliable data (controlled conditions and multiple repetitions) for comparison between the evaluated systems. Moreover, although some studies have already been conducted [2,4,35], information about the nutrient requirements of insects and the evaluation of alternative by-products as insect diets are still needed [3]. Experimental rearing substrates are by-products characterized by a wide variability in chemical composition, which affected growth performance and the chemical composition of larvae. This is particularly important, since the growth of insect-based products on the market should consider insect diet, processing conditions, and product properties, with insect diet being the most critical point for environmental impact [21]. In the present study, larval mortality was very low and in the range reported by several authors on a wide variety of rearing substrates [36,37]. According to Cammack and Tomberlin [38], larvae reared on a balanced diet of protein (21%) and carbohydrates (21%) developed the fastest on the least amount of food and had the greatest survival rate. Similarly, Tschirner and Simon [4] showed that the nutrient composition of rearing substrates has a great influence on critical production factors like total larvae yield and individual larvae body weight, with the best production results achieved with a mixture of middlings (CP 22% on DM), whereas the use of dry sugar beets and high distiller grains led to more unfavourable results. However, as shown by Meneguz et al. [3], BSF larvae are able to bioconvert waste and by-products characterized by high fiber content for the presence of cellulase enzymes [39]. Lee et al. [40] also reported the presence of a cellulase within the gut microbiota of *H. illucens.* In the present study, BSF larvae were able to grow on by-products characterized by a high NDF content, such as brewer’s grains, although with a low growth rate and final larval weight, which can be a disadvantage for industrial scale production. Other studies [3,37] reared BSF larvae on high fiber diets (brewery by-products), obtaining similar results in terms of larval mortality, length of larval development, and bioreduction. The results obtained by ur Rehman et al. [41] on okara were similar to those obtained in this paper. Overall, in the present study, BSF larvae reared on diets with the highest content of NFC (hen and maize distillers) were characterized by the highest weight. 

During performance trials, the potential emission of CH_4_ by BSF larvae was evaluated. In insects, CH_4_ can be produced from bacterial fermentation by methanobacteriaceae in the hindgut [42]. However, very few measurements of methane (and other GHGs) have been gathered from insects. The results of the present study show that BSF larvae did not produce any detectable trace of CH_4_, similar to the results of Perednia et al. [43]. The low/zero emissions likely occur because insects are mainly fed a high protein-low, fiber diet, and, for this reason, they do not use microbes to break down dietary cellulose [19]. However, as hypothesized by Halloran et al. [19], in the future, diets containing cellulose, hemicellulose, and complex lignocellulose compounds will be used as feed for insects, and it is, therefore, likely that methane emissions may become a problem. However, it must be underlined that in this study the high fiber diet (brewer’s grain) did not affect CH_4_ production.

Considering the effects of the rearing substrate on the larvae’s chemical composition, the CP content of larvae was higher than that reported by Sánchez-Muros et al. [44]. All the diets tested in the present study were characterized by considerably high CP content, which, in turn, could have positively affected the CP content of larvae. The protein and fat compositions of BSF depend on the diet. As reported by Wang and Shelomi [45], natural variation among individuals and productive batches affect the CP and EE contents of BSF larvae with values ranging from 31.7% to 47.6% for crude protein and 11.8%–34.3% for fat. The EE content of larvae was significantly affected by diets with higher values for larvae reared on okara and distillers, the substrates with the highest EE concentrations. This agrees with the results of Spranghers et al. [15], which showed that the EE and ash contents of BSF were significantly affected by the EE content of the rearing substrate. 

The NPK content of the feed refusals and manure produced showed a similar proportion among nutrients as livestock manures with a high nitrogen content and low phosphorous content, but this result was slightly different than the data reported by Thevenot et al. [46]. The composition was strictly influenced by substrate composition and larvae utilization. The highest percentage of nitrogen in the refusal was found for larvae grown on maize distillers, which can be explained by the low WRI of this diet. On the other hand, the lowest percentage of nitrogen showed by brewer’s grain refusals was due to the reduced protein content of the substrate.

Considering the environmental impact, as suggested by van Huis and Oonincx [47], the high feed conversion efficiency of the insects is one of the main reasons why insects are considered to be potentially sustainable sources of protein. Feed conversion efficiency is also the main driver that influences the environmental sustainability of animal production [48,49,50]. However, as underlined by Smetana et al. [20], a higher efficiency in term of insect yield (and protein content) is usually achieved using feed with good nutritional quality (e.g., rye meal, soybean meal), but in this case, insect production is usually associated with high environmental impacts. This was confirmed by the results of the present study, with larvae grown on a hen diet having, for example, a greater contribution to climate change. The climate change assessment value for larvae production on a hen diet was higher than the results found by Thevenot et al. [46], with larvae grown on a cereal by-product meal. The composition of the diet for insect rearing was different, and this played an important role in the final result. According to Smetana et al. [20], the most promising feeds for insects are those based on low value agri-food products with good nutritional profiles, such as distiller dry grains, as demonstrated by “in vivo performance” trials conducted in the present study. 

Using different FUs to express the environmental impact of larvae production is useful for comparing the results with the benchmark feeds. In the present study, the production of 1 kg of protein from insects was influenced by the rearing substrate, and a lower climate change effect could be achieved when the insect rearing occurred on low impact feeds. From the results of the present study, the climate change for the production of 1 kg of protein from the BSF reared on a maize distiller was comparable to the impact generated by the production of protein from a local (European) feed as a sunflower meal. The use of by-products for insect production would enable one to re-use products, thereby encouraging a circular economy. Moreover, protein production from insects could increase the production of local proteins, avoiding, for example, the import of huge quantity of soybean meal for European animal feeding. 

The comparison with fish meal showed that the environmental impact of producing one kilo of protein is lower for fish meal than for BSF, which can reduce the interest in BSF as an alternative protein source to fish meal. In any case, the use of insect meal in fish feeding can provide a positive alternative for the depletion of fish stock, as reported by Thevenot et al. [46].

As previously described, substrates were characterized by different EE content, affecting the EE content of larvae, especially of larvae reared on a hen diet and on brewer’s grain. These results yielded a great load on the environmental impact values per kg of lipid (Figure 5). Other authors [21] found that, when compared with an alternative source of protein such as soymeal or an alternative source of lipids such as rapeseed, the BSF system yielded a higher climate change value and a higher eutrophication.

The relative weight of the different processes of larvae grown on hen diets and maize distiller (Figure 2 and Figure 3) feed production activities were the main contributors to the environmental impact. Because of the high impact of feed production, assigning an environmental load to substrate production using sensitive analyses, especially when the substrate is a by-product, was a priority of this study. These analyses made a comparison among different production of larvae possible. In this way, the most sustainable product was the larvae grown on the maize distiller, which showed the lowest value for the most impact categories considered. The results obtained from the sensitivity analysis underline that, for the future, given the growing importance attributed to the inclusion of food by-products in a livestock production system, it is necessary to define a methodology to assign the environmental load of these materials.

## 5. Conclusions

The selected by-products resulted useful to be included in insect breeding system for their easily availability and low cost and interesting in circular economy approach. 

A careful selection of rearing substrate is very important for optimizing the growth, quality, and environmental impact of the production of insects for animal nutrition. All the substrates tested in the present study were characterized by a high quality in terms of CP and EE content (particularly for okara and maize distillers), which allowed a high conversion efficiency and BSF larval growth. Since the composition of substrates affected the quality of larvae, further studies are needed to evaluate the diet requirements of BSF in relation to the quality of products.

Although all the substrates promote the growth of larvae, the maize distiller seems the most promising substrate in terms of environmental sustainability.

Considering that substrate production importantly contributes to the environmental impact of larvae production, it is necessary to deepen the knowledge and methods of substrate production to attribute an environmental weight to by-products. Only by considering this crucial point will it be possible to better evaluate the environmental benefits of producing insects as novel feed and/or food. 

## Figures and Tables

**Figure 1 animals-09-00289-f001:**
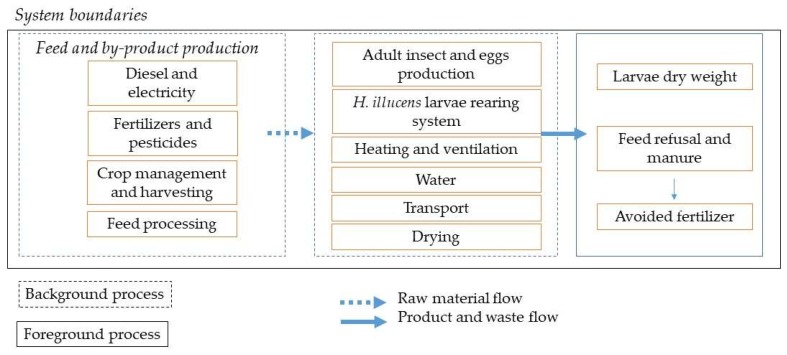
System boundaries of the production process considered.

**Figure 2 animals-09-00289-f002:**
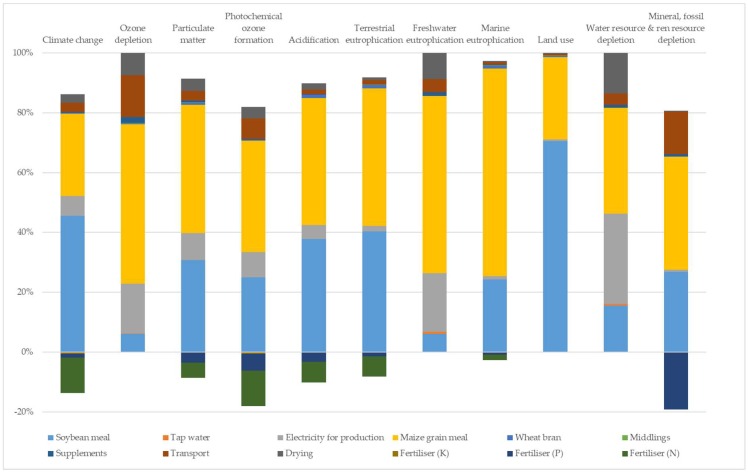
Relative weight of different processes of larvae grown on a hen diet.

**Figure 3 animals-09-00289-f003:**
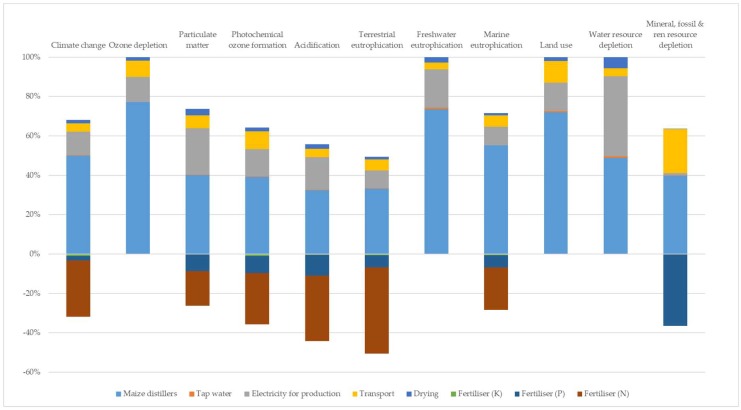
Relative weight of different processes of larvae grown on maize distiller.

**Figure 4 animals-09-00289-f004:**
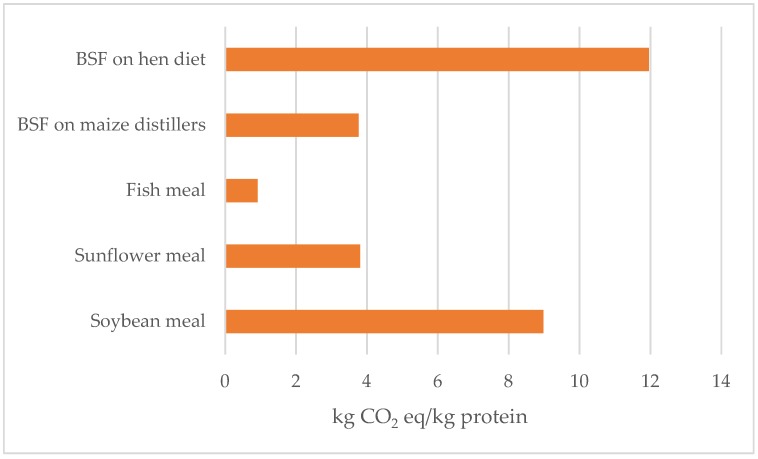
Climate change estimation to produce 1 kg of protein from black soldier fly (BSF) larvae production (on different substrates) from fish meal and from two vegetable feeds (sunflower meal and soybean meal).

**Figure 5 animals-09-00289-f005:**
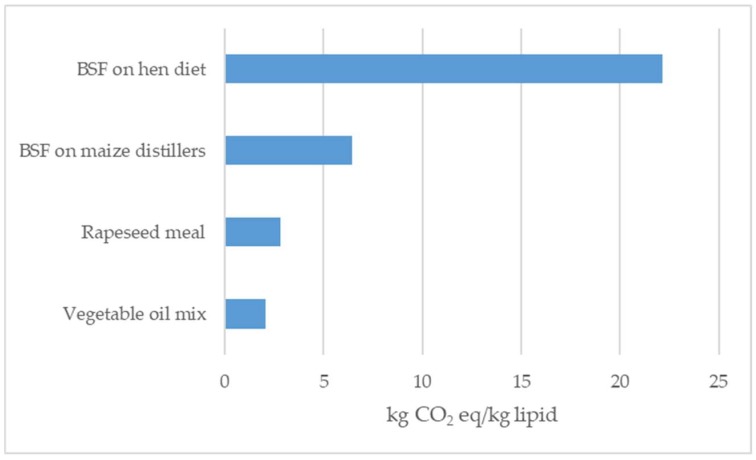
Climate change estimation to produce 1 kg of lipids from BSF larvae production (on different substrates) from two types of vegetable feed (a rapeseed meal and vegetable oil mix).

**Figure 6 animals-09-00289-f006:**
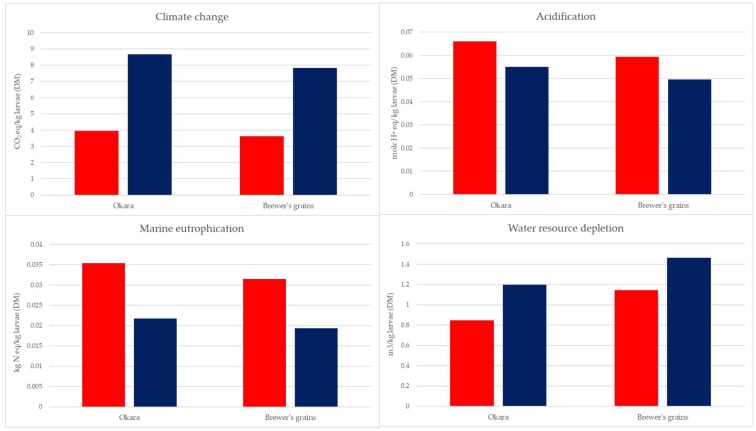
Sensitivity analysis: environmental impact evaluation substituting okara and brewer’s grain with sunflower meal (red bar) or soybean meal (blue bar) based on protein content.

**Table 1 animals-09-00289-t001:** Main items of life cycle inventory for the larvae production (1 kg dry matter (DM)).

	Hen Diet	Maize Distiller	Okara	Brewer’s Grains
Input				
Ingested feed, kg of DM	4.22	2.81	2.80	3.30
Electricity, kWh	0.81	1.00	1.40	2.08
Tap water, kg	9.61	9.78	13.7	0
Transport, lorry 16–32 t, km	200	200	300	100
Drying, 60 °C for 24 h, kWh	0.363	0.1411	0.1410	0.1413
Output				
Larvae production, kg of DM	1	1	1	1
Protein production from larvae, kg protein	0.48	0.51	0.50	0.53
Lipid production from larvae, kg lipids	0.26	0.30	0.32	0.25
Feed refusal and manure, kg of DM	3.056	2.757	0.583	0.850

**Table 2 animals-09-00289-t002:** Dry matter, ash, crude protein (CP), ether extract (EE), neutral detergent fiber (NDF) and non-fibrous carbohydrate (NFC) concentrations (% on DM) of the experimental substrates unless stated differently.

Experimental Substrates	Dry Matter% on A Fed Basis	Ash	CP	EE	NDF	NFC
Hen diet	92.1	13.5	17.0	4.00	15.7	49.8
Okara	18.3	4.13	39.2	17.2	32.0	7.47
Maize distillers	94.9	5.40	29.5	11.1	36.7	17.3
Brewer’s grains	15.8	4.13	15.8	2.89	53.6	11.2

**Table 3 animals-09-00289-t003:** Performance of larval growth and bioreduction of the substrates.

Substrate	Larval Weight (G) (*n* = 10) (Fresh Weight)	Larval Survival (%)	WRI	ECD	GR
Hen diet	2.29 ± 0.20 b	97.53 ± 1.86 a	4.46 ± 0.36 b	0.27 ± 0.02 a	0.0051 ± 0.0007 b
Maize Distillers	1.97 ± 0.14 b	73.00 ± 11.92 a	3.22 ± 0.21 a	0.27 ± 0.02 a	0.0056 ± 0.0001 b
Okara	1.38 ± 0.06 a	98.5 ± 0.84 a	4.90 ± 0.07 b	0.36 ± 0.02 b	0.0021 ± 0.0.000 a
Brewer’s grain	0.98 ± 0.01 a	95.87 ± 1.51 a	3.01 ± 0.06 a	0.25 ± 0.01 a	0.0014 ± 0.0000 a

WRI = Waste reduction index; ECD = Efficiency of conversion of the ingested food; GR= growth rate index (g d-1). Means within a column with different letters differ (*p* < 0.05)

**Table 4 animals-09-00289-t004:** Dry matter (DM, %), ash, crude protein (CP) and ether extract (EE) concentrations (% on DM) of the larvae grown on the experimental substrates.

Experimental Substrates	DM	Ash	CP	EE
Hen diet	38.9	11.7	52.8	25.1
Okara	37.4	5.91	51.2	31.2
Maize distillers	38.5	4.94	53.4	29.9
Brewer’s grains	36.5	7.41	54.1	23.2

**Table 5 animals-09-00289-t005:** Chemical analysis of feed refusal and manure.

Chemical Analysis	Hen Diet	Maize Distillers	Okara	Brewer’s Grains
N content in feed refusal and manure, %	3.35	4.94	3.52	3.26
K content in feed refusal and manure, %	1.93	1.92	2.70	0.11
P content in feed refusal and manure, %	1.26	1.32	0.88	0.88

**Table 6 animals-09-00289-t006:** Environmental impact of larvae grown on different substrates (kg of larvae dry weight).

Impact Category	Unit	Hen Diet	Maize Distillers	Okara	Brewer’s Grains
Climate change	kg CO_2_ eq	5.76	1.95	0.68	0.81
Ozone depletion	g CFC^−11^ eq	2.76 × 10^−7^	4.47 × 10^−7^	9.61 × 10^−8^	1.41 × 10^−7^
Particulate matter	g PM^2.5^ eq	1.66	0.45	0.31	0.42
Photochemical ozone formation	g NMVOC eq	10.2	3.39	1.91	2.38
Acidification	molc H^+^ eq	0.049	0.002	0.004	0.004
Terrestrial eutrophication	molc N eq	0.205	−0.001	0.003	0.001
Freshwater eutrophication	g P eq	0.50	0.61	0.20	0.28
Marine eutrophication	g N eq	37.7	2.43	0.55	0.64
Land use	kg C deficit	94.7	4.92	1.25	1.79
Water resource depletion	m^3^ water eq	1.26	1.16	0.75	1.06
Mineral, fossil, and ren resource depletion	kg Sb eq	2.95 × 10^−5^	8.06 × 10^−6^	−1.34 × 10^−7^	−1.17 × 10^−6^

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
