# Peer review of "Rearing of Hermetia Illucens on Different Organic By-Products: Influence on Growth, Waste Reduction, and Environmental Impact"

_animals, 2019, doi:10.3390/ani9060289_

Round 1

Reviewer 1 Report

I would congratulate with authors because the paper is very  well presented, data are scientific sounds, M&M described in detail, the discussion provide original insight. The only concern is referred to the poor number of larvae/replicates (N=10) weighed at each control. Some paper are available testing the use of different substrates on larvae growth performance, could you underline the innovative approach of  your paper testing different substrates? so how does the paper stand out from others in this field?

Could you justify how did you chise to weight 10 larvae/replicate? which is the rationale?

The paper is very rich in terms of information related to  LCA. I recommend the publication of the paper in the present  form, I only suggest to provide some numerical data in  brackets in the abstract.

Author Response

AU: The authors thank the reviewer for the precious revision and suggestions to improve the paper. We accept all the indications and modified the text in the revision form.

Review 1

Open Review

English language and style

( ) Extensive editing of English language and style required 
( ) Moderate English changes required 
( ) English language and style are fine/minor spell check required 
(x) I don't feel qualified to judge about the English language and style 

Yes

Can be improved

Must be improved

Not applicable

Does the introduction provide   sufficient background and include all relevant references?

(x)

( )

( )

( )

Is the research design appropriate?

(x)

( )

( )

( )

Are the methods adequately described?

(x)

( )

( )

( )

Are the results clearly presented?

(x)

( )

( )

( )

Are the conclusions supported by the   results?

(x)

( )

( )

( )

Comments and Suggestions for Authors

I would congratulate with authors because the paper is very well presented, data are scientific sounds, M&M described in detail, the discussion provide original insight. The only concern is referred to the poor number of larvae/replicates (N=10) weighed at each control. Some paper are available testing the use of different substrates on larvae growth performance, could you underline the innovative approach of your paper testing different substrates? so how does the paper stand out from others in this field? 

AU: We added some explanation in the aim and the conclusion section

Could you justify how did you chise to weight 10 larvae/replicate? which is the rationale?

 AU: 10 larvae for each replicate were used only to define the growth trend on the different diets. In other papers, Authors weighted a different number of larvae to obtain the growth trend (Nguyen et al. (2015) and Tinder et al. (2017) weighed 3 larvae a time, while Lalander et al (20198) 10 larvae, and Meneguz et al. (2018) 30. Thus, we think 10 larvae are adequate for the purpose. Instead, to calculate different biodegradation indexes all larvae were weighted and counted.     

The paper is very rich in terms of information related to LCA. I recommend the publication of the paper in the present form, I only suggest to provide some numerical data in brackets in the abstract

AU: We added some details in the abstract

Reviewer 2 Report

- Overall, there are numerous grammatical errors that while minor on their own, make reading the manuscript difficult at times. 

- the background materials and base model information used to define environmental impacts need much further clarification and discussion. 

- Please clarify water addition. Methods say 1:1 for hen diet and maize, bt Table 1 show much greater ratio of water:feedstock. 

- Table 6 is very difficult to read- font size needs to be uniform and column width adjusted for clarity

- Each of the factors in Table 6 needs to be defined and the basis for the calculation defined as well

- the basis and background for discussion of impacts of fishmeal, sunflower meal need to be clarified - where are the data coming from? 

Author Response

AU: The authors thank the reviewer for the precious revision and suggestions to improve the paper. We accept all the indications and modified the text in the revision form.

Open Review

English language and style

( ) Extensive editing of English language and style required 
(x) Moderate English changes required 
( ) English language and style are fine/minor spell check required 
( ) I don't feel qualified to judge about the English language and style 

Yes

Can be improved

Must be improved

Not applicable

Does the introduction provide   sufficient background and include all relevant references?

(x)

( )

( )

( )

Is the research design appropriate?

( )

(x)

( )

( )

Are the methods adequately described?

( )

( )

(x)

( )

Are the results clearly presented?

( )

(x)

( )

( )

Are the conclusions supported by the   results?

( )

( )

(x)

( )

Comments and Suggestions for Authors

- Overall, there are numerous grammatical errors that while minor on their own, make reading the manuscript difficult at times. 

AU: We followed this suggestion and we improved the text

- the background materials and base model information used to define environmental impacts need much further clarification and discussion. 

AU: We added some details of background materials in Materials and Methods section

- Please clarify water addition. Methods say 1:1 for hen diet and maize, bt Table 1 show much greater ratio of water:feedstock. 

AU: We modified the water values in table 1. The ratio 1:1 is referred to the total quantity of by-products administrated: water at the beginning of the breeding period. In the table 1 we reported the total feed intake (administrated minus residues) and total amount of water consumed

- Table 6 is very difficult to read- font size needs to be uniform and column width adjusted for clarity

AU: We corrected the table

- Each of the factors in Table 6 needs to be defined and the basis for the calculation defined as well

AU: We modified the tables and correct the results after the correction of water quantity reported in table 1

AU: We added some details about chosen method

- the basis and background for discussion of impacts of fishmeal, sunflower meal need to be clarified - where are the data coming from?

AU: We added some details in Materials and Methods section